# A Clinical Risk Model for Personalized Screening and Prevention of Breast Cancer

**DOI:** 10.3390/cancers15123246

**Published:** 2023-06-19

**Authors:** Mikael Eriksson, Kamila Czene, Celine Vachon, Emily F. Conant, Per Hall

**Affiliations:** 1Department of Medical Epidemiology and Biostatistics, Karolinska Institutet, 171 65 Stockholm, Sweden; 2Centre for Cancer Genetic Epidemiology, Department of Public Health and Primary Care, University of Cambridge, Cambridge CB1 8RN, UK; 3Division of Epidemiology, Department of Quantitative Health Sciences, Mayo Clinic College of Medicine, Rochester, MN 55905, USA; 4Department of Radiology, Perelman School of Medicine, University of Pennsylvania, Philadelphia, PA 19104, USA; 5Department of Oncology, Södersjukhuset University Hospital, 118 83 Stockholm, Sweden

**Keywords:** breast cancer, risk model, long-term risk, primary prevention, individualized screening, artificial intelligence, image-derived risk model

## Abstract

**Simple Summary:**

We investigated the benefits of adding lifestyle and familial risk factors to a mammographic image-derived short-term AI risk model in a 10-year follow-up study for its potential use in personalized screening and prevention of breast cancer (BC). In a case–cohort study, 8110 women were selected from women aged 40–74 participating in a Swedish mammography screening cohort. The women had no BC diagnosis at enrollment. In all, 1661 incident BCs were developed in the case–cohort. The lifestyle/familial-expanded AI risk model showed a significantly higher discriminatory performance in the long term and short term than the imaging-only risk model and the clinical Tyrer–Cuzick v8 model. The expanded model also showed the highest risk classification performance using positive predictive value (PPV). The results suggest that a lifestyle/familial-expanded image-derived AI risk model could most efficiently refine the identification of women who may benefit from personalized screening and/or risk-reducing intervention.

**Abstract:**

Background: Image-derived artificial intelligence (AI) risk models have shown promise in identifying high-risk women in the short term. The long-term performance of image-derived risk models expanded with clinical factors has not been investigated. Methods: We performed a case–cohort study of 8110 women aged 40–74 randomly selected from a Swedish mammography screening cohort initiated in 2010 together with 1661 incident BCs diagnosed before January 2022. The imaging-only AI risk model extracted mammographic features and age at screening. Additional lifestyle/familial risk factors were incorporated into the lifestyle/familial-expanded AI model. Absolute risks were calculated using the two models and the clinical Tyrer–Cuzick v8 model. Age-adjusted model performances were compared across the 10-year follow-up. Results: The AUCs of the lifestyle/familial-expanded AI risk model ranged from 0.75 (95%CI: 0.70–0.80) to 0.68 (95%CI: 0.66–0.69) 1–10 years after study entry. Corresponding AUCs were 0.72 (95%CI: 0.66–0.78) to 0.65 (95%CI: 0.63–0.66) for the imaging-only model and 0.62 (95%CI: 0.55–0.68) to 0.60 (95%CI: 0.58–0.61) for Tyrer–Cuzick v8. The increased performances were observed in multiple risk subgroups and cancer subtypes. Among the 5% of women at highest risk, the PPV was 5.8% using the lifestyle/familial-expanded model compared with 5.3% using the imaging-only model, *p* < 0.01, and 4.6% for Tyrer–Cuzick, *p* < 0.01. Conclusions: The lifestyle/familial-expanded AI risk model showed higher performance for both long-term and short-term risk assessment compared with imaging-only and Tyrer–Cuzick models.

## 1. Introduction

A key step in efficient personalized breast cancer screening and risk-reducing interventions is improving breast cancer risk assessment [1,2]. Recent studies have reported that artificial intelligence (AI)-based models using mammograms perform better than traditional lifestyle familial-based risk models in estimating the short-term risk of breast cancer [3,4,5]. The long-term performance of image-derived models that incorporate lifestyle and family history risk factors has yet to be determined.

Risk assessment for breast cancer is currently offered in the U.S. using lifestyle/familial-based risk tools such as the Tyrer–Cuzick v8 and Gail models [6,7,8]. Women at a high risk of breast cancer are recommended to undergo supplemental screening with modalities such as breast MRI to increase the early detection of breast cancer. In addition, women at high risk of developing breast cancer may benefit from risk reduction interventions to decrease their risk of developing breast cancer [9,10,11].

Primary prevention efforts are important because approximately 13% of women are currently diagnosed with breast cancer over their lifetime [12]. Lifestyle changes and medical interventions have shown promising results for reducing breast cancer incidence in high-risk women [13]. Clinical guidelines for high-risk women are also available such as the National Institute for Health and Care Excellence (NICE) and the US Preventive Services Task Force (USPSTF) [14,15].

We used the KARolinska MAmmography Project for Risk Prediction of Breast Cancer (KARMA) screening cohort [16] to investigate the long-term predictive performance using an image-derived AI-based risk model that was expanded with lifestyle risk factors and family history of breast cancer. In this long-term evaluation study, we used an independent set of baseline mammograms from the same underlying screening cohort that was used to develop the model [17]. The AI risk models are available for clinical use in the U.S. and Europe [18].

We studied the predictive performances up to 10 years after study entry to identify women who may benefit from risk-reducing intervention using the lifestyle/familial-expanded model compared to the imaging-only AI risk model, which in turn was compared to the clinically used Tyrer–Cuzick v8 risk model, which includes mammographic density. We also investigated the three models for short-term risk to identify women who may benefit from supplemental screening or a shorter screening interval. We performed analyses in the overall study population and stratified by the risk factors included in the Tyrer–Cuzick v8 risk model to investigate the generalizability of the models to populations with differential risk factor distributions as well as to investigate subgroups of women who could benefit from using the Tyrer–Cuzick v8 model. In addition, we compared the risk models for their ability to capture women who will develop subtype-specific breast cancers.

## 2. Materials and Methods

### 2.1. Study Population

In the Swedish national mammography screening program, women aged 40–74 years are invited every 18 or 24 months, depending on age and region [19]. Women who underwent mammographic screening at four hospitals between October 2010 and March 2013 were invited to participate in the prospective KARMA study [16]. Seventy thousand women consented to participate in research on the risk of breast cancer and responded to a web survey on lifestyle and familial-related risk factors for breast cancer. In addition, women donated blood, approved linkages to national registers, and allowed storage and image analysis of their mammograms. In the present study, women with no diagnosis of breast cancer at study entry were eligible. A case–cohort study was conducted consisting of a random subcohort of 8110 (12.1%) from the 66,814 eligible KARMA women together with all incident breast cancers with available risk information that were diagnosed before the register linkage, where 197 of the incident breast cancers were in the subcohort. The article followed the Strengthening the Reporting of Observational Studies in Epidemiology (STROBE) reporting guidelines for cohort studies.

The study was approved by the Swedish Ethical Review Authority (2010/958-31/1).

### 2.2. Risk Factors at Enrollment and Breast Cancers at Follow-Up

Full-field digital mammographic (FFDM) images were obtained from the left and right breasts (mediolateral oblique (MLO) and cranio-caudal (CC) views) and used to extract AI-based mammographic features (density, microcalcifications, masses, and left–right breast asymmetries of these features) using the ProFound AI Risk (iCAD, Nashua, NH) imaging-only risk tool as previously described (Model 1) [17,20].

The lifestyle/familial-expanded AI risk model used Body Mass Index (BMI), at least monthly use of alcohol, regular smoking, current use of hormone replacement therapy (HRT), and first degree family history of breast cancer in addition to the image-derived features and age as previously described [17]. The factors were extracted from the questionnaire developed as part of the KARMA project [16]. The lifestyle/familial-expanded AI risk model was constructed using the mammographic features in the imaging-only risk model in combination with the lifestyle/familial factors as previously described (Model 2) [17].

The Tyrer–Cuzick v8 risk model was used as a clinical comparison tool [21]. Tyrer–Cuzick v8 provides breast cancer risk based on lifestyle and familial risk factors, i.e., age, height, weight, age at menarche, age at first childbirth, menopausal status, use of hormone replacement therapy (HRT), previous benign breast disorders, first-, second-, and third-degree family history of breast cancer, first- and second-degree family history of ovarian breast cancer, and mammographic density [20]. In our analysis, we did not have access to information regarding history of ovarian cancer in second-degree relatives. We classified mammographic density using the fully automated mammographic density tool STRATUS [20]. Mammographic density was classified into four cBIRADS categories to mimic the Breast Imaging Reporting and Data System (BI-RADS) classification as previously described [22].

Absolute risks of breast cancer were calculated for the three models based on their inclusion of risk factors, risk factor prevalence, and Swedish national statistics on breast cancer incidence rate and competing mortality risk [23]. Risk assessment was performed at study enrollment.

Breast cancer status and mode of detection were retrieved for breast cancers diagnosed up to January 2022 from the National Quality register for Breast Cancer (NKBC) register through linkage using the unique Swedish personal identification numbers [24]. Symptomatic cancer was defined by mode of detection as non-screen-detected cancer. Tumor characteristics were defined using the American Joint Committee on Cancer (AJCC) classification [25].

### 2.3. Statistical Analysis

Descriptive statistics reported study participant characteristics at study entry [26]. The frequency distribution reported the time from date of mammogram at enrollment to date of breast cancer diagnosis. Absolute risks were estimated at study entry using the three risk models. Inverse probability weights were used to account for the case–cohort sampling [26,27]. Area under the receiver operating characteristics curve (AUC) estimated the discriminatory performance of the models across the 1–10-year follow-up period after age adjustment [28,29]. To estimate the discriminatory performance for a certain year of follow-up, we used the subcohort and the incident breast cancers that developed after study entry between 3 months and that year. The 95% confidence intervals of the AUC point estimates were estimated using 1000 bootstraps [29]. In a subgroup analysis, we also excluded breast cancers diagnosed in the first two years after study baseline. Differences in performances between the lifestyle/familial-expanded risk model and the imaging-only risk model, and in turn between the imaging-only risk model and Tyrer–Cuzick v8, were tested using bootstrapping [30]. The discriminatory performances were reported first for the overall study population and then for subgroups of women stratified by the risk factors included in Tyrer–Cuzick and by breast cancer subtypes (mode of detection, invasiveness, stage, and receptor status). The lifestyle/familial-expanded AI risk model and Tyrer–Cuzick imputed risk scores for missing risk factor data using built-in methods [17,21]. Manhattan plots and bubble plots presented the significance of risk model AUC differences in the subgroup analyses after adjusting for multiple testing using the Holm–Bonferroni method [31,32,33]. Positive Predictive Values (PPVs) were estimated with 95% confidence intervals [34]. The proportions of women at low, general, moderate, and high risk based on two-year and ten-year risks were reported according to the National Institute for Health and Care Excellence (NICE) and the U.S. Preventive Service Task Force (USPSTF) guidelines.

Statistical analyses were performed using R 4.1 [35]. All tests were two-sided at significance level 0.05.

## 3. Results

### 3.1. Study Population

The case–cohort of 9574 women consisted of 8110 women in the subcohort and 1661 incident breast cancers that were identified in the January 2022 register update as being diagnosed more than three months after enrollment in 2011–2013. A total of 197 out of the 1661 incident breast cancers were found in the subcohort, as shown in Table 1. At study entry, the mean age was 56.52 (±9.53) in cases and 53.86 years (±9.85) in the subcohort. The absolute 2-year risk at baseline was 1.37%, 1.19%, and 0.79% in cases using the lifestyle/familial-expanded AI risk model, the imaging-only AI risk model, and the Tyrer–Cuzick v8 risk model, respectively. The corresponding absolute risks were 0.66%, 0.63%, and 0.60%.

### 3.2. Follow-Up of Breast Cancers from Time of Enrollment

The follow-up period for the women in this study was over 10 years and the time from the mammogram at enrollment to breast cancer diagnosis ranged from 3 months to over 10 years, as shown in Figure A1. Half of the cancer events occurred in the first five years. After 10 years of follow-up, 60% of the breast cancers were screen-detected, 31% were diagnosed with a stage 2 or later cancer, 86% were invasive, 86% were estrogen-receptor (ER)-positive, 72% were progesterone-receptor (PR)-positive, and 14% were human epidermal growth factor receptor 2 (HER2)-positive, as shown in Table A1.

### 3.3. Discriminatory Performance Overall and in Subgroups

The age adjusted discriminatory performance ranged from 0.75 (95%CI 0.70–0.80) after 1 year of follow-up to 0.68 (95%CI 0.66–0.69) after 10 years for the lifestyle/familial-expanded AI risk model, as shown in Figure 1, Table A2. The corresponding 1- and 10-year AUCs were 0.72 (95%CI 0.66–0.78) and 0.65 (95%CI 0.63–0.66) for the imaging-only AI risk model and 0.62 (95%CI 0.55–0.68) and 0.60 (95%CI 0.58–0.61) for Tyrer–Cuzick v8.

The AUC point estimates for the lifestyle/familial-expanded AI risk model ranged from 0.75 (95%CI 0.70–0.80) after 1 year of follow-up to 0.68 (95%CI 0.66–0.69) after 10 years of follow-up. The corresponding estimates for the imaging-only AI risk model were 0.72 (95%CI 0.66–0.78) and 0.65 (95%CI 0.63–0.66), and for the Tyrer–Cuzick v8 model, they were 0.62 (95%CI 0.55–0.68) and 0.60 (95%CI 0.58–0.61). The horizontal line denotes the AUC point estimate (0.68) of the lifestyle/familial-expanded AI risk model for 10 years of follow-up. When excluding breast cancers diagnosed in the first two years after study baseline, we observed AUC point estimates for the lifestyle/familial-expanded AI risk model ranging from 0.72 (95%CI 0.67–0.77) after 3 years of follow-up to 0.65 (95%CI 0.61–0.69) after 10 years of follow-up, as shown in Figure A2. The corresponding estimates for the imaging-only AI risk model were 0.70 (95%CI 0.65–0.75) and 0.61 (95%CI 0.57–0.65), and for the Tyrer–Cuzick v8 model, they were 0.63 (95%CI 0.54–0.69) and 0.60 (95%CI 0.58–0.62).

Discriminatory performance tests (N = 160) were performed on risk factor subgroups of women comparing the lifestyle/familial-expanded AI risk model and Tyrer–Cuzick v8 with the imaging-only AI risk model, as shown in Figure 2. The number of subgroups with a significantly different AUC and the strength of significance increased with the number of years of follow-up using the lifestyle/familial-expanded AI risk model, while the corresponding AUCs using Tyrer–Cuzick v8 were similar across the 1–10-year follow-up (Figure 2). Using the lifestyle/familial-expanded AI risk model, all 86 comparisons that were significantly different had a higher AUC, whereas using the Tyrer–Cuzick v8 risk model, all 52 comparisons with a significant difference AUC were lower (Figure A3).

The lifestyle/familial-expanded AI risk model showed significantly higher AUCs compared to the imaging-only AI risk model after 2 years of follow-up, when women with at least monthly intake of alcohol, postmenopausal women, women above median age, below median length, below median weight, below median BMI, above median age at menarche, and above median age at first childbirth benefitted from being assessed with the lifestyle/familial-expanded model compared to the imaging-only AI risk model (Figure A3 bubble plot). After 10 years of follow-up, all nine risk factors (alcohol intake, menopause, age, height, weight, BMI, age at menarche, age at first childbirth, benign breast disease) were significant. Women with a family history of breast cancer did not have a significantly increased AUC over the imaging-only AI risk model, nor did women with second/third-degree breast cancer family history or ovarian cancer in the family (Figure A3).

### 3.4. Predictive Value by Tumor Characteristics

The age adjusted discriminatory performance estimations by breast cancer subtypes resulted in 64 tests comparing the lifestyle/familial-expanded AI risk model with the imaging-only AI risk model, which in turn was compared with Tyrer–Cuzick v8 (Figure A4). Similar tendencies were observed with increasing number of significant AUC differences and increasing years of follow-up for the lifestyle/familial-expanded model, while AUC differences were not depending on time for Tyrer–Cuzick v8. All 36 significant comparisons for the lifestyle/familial-expanded model showed higher AUCs, in contrast to all 36 significant comparisons for the Tyrer–Cuzick v8 risk model, which showed lower AUCs (Figure A5).

The significant discriminatory improvements of the lifestyle/familial-expanded AI risk model over the imaging-only AI risk model ranged from 2.3–4.3 percentage points for screen-detected, symptomatic, invasive, ER-, PR-, HER2-positive, and stage 2 or later breast cancers (Figure 3, Table A3). Screen-detected, invasive, ER-, and PR-positive breast cancers were significant from 2 years of follow-up, while stage 2 or later cancers and HER2 cancers were significant after 8–10 years of follow-up. For Tyrer–Cuzick v8, screen-detected, invasive, in situ, ER- and PR-positive, and stage 2 or later breast cancers had significantly lower AUCs (3.7–25.8 percentage points) compared with the imaging-only model at one or several years in the 1–10-year follow-up (Figure 3, Table A4).

### 3.5. Positive Predictive Value (PPV) and Clinical Risk Classification

PPVs were estimated in multiple subgroups of women ranging from 1% to 20% of women at highest risk of breast cancer for each of the three models. Using the lifestyle/familial-expanded AI risk model, the PPVs ranged from 4.8% (95%CI 4.7–5.0) to 4.7% (95%CI 4.5–4.8) across the 1–20% of women at the highest risk (Figure 4). The corresponding PPVs for the imaging-only AI risk model were 4.6% (95%CI 4.5–4.8) and 4.2% (95%CI 4.1–4.4) and for Tyrer–Cuzick v8, they were 4.0% (95%CI 3.9–4.2) and 4.0% (95%CI 3.8–4.1). The 5% of women at highest risk had a PPV of 5.8% for the lifestyle/familial-expanded AI risk model compared with 5.3% for the imaging-only AI risk model, *p* < 0.01, and 4.6% for Tyrer–Cuzick v8, *p* < 0.01.

The proportion of breast cancers that were identified as high-risk per NICE guidelines using the lifestyle/familial-expanded AI risk model (22%) was significantly larger than the corresponding proportions identified using the imaging-only AI risk model (19%), *p* < 0.01, and Tyrer–Cuzick v8 (7.2%) with 10-year risk projection, *p* < 0.01 (Table A5). The high-risk women were 6.7 times more likely to be diagnosed with breast cancer compared to women at general risk using the lifestyle/familial-expanded model (Table A5). The corresponding figure for the imaging-only model was 6.4 and, for Tyrer–Cuzick v8, 4.2. Similar differences in risk stratification performances were observed using USPSTF guidelines (Table A6).

The positive predictive value is presented after inverse probability weighting to account for the case–cohort sampling with 95% confidence intervals.

Tests were performed to determine the difference between the binomial proportions of breast cancers captured at high risk by the lifestyle/familial-expanded AI risk model, the imaging-only AI risk model, and Tyrer–Cuzick v8. In the 5% of women at the highest risk, the proportion of cancer that was captured by the lifestyle/familial-expanded AI risk model (5.8%) was significantly higher than the 5.3% captured by the image-based risk model, *p* < 0.01, which in turn was significantly higher than the 4.6% of breast cancers that were captured by Tyrer–Cuzick v8 10-year, *p* < 0.01. The corresponding numbers for the 12% of women at highest risk were 5.3 and 4.9, *p* < 0.01, and 4.5, *p* < 0.01.

## 4. Discussion

We investigated the performance of a lifestyle/familial-expanded AI risk model over an imaging-only-derived AI risk model, in turn, over the clinical Tyrer–Cuzick v8 risk model within a large-scale prospective screening cohort. The lifestyle/familial-expanded AI risk model showed significantly better discriminatory performances of 2–4% over the imaging-only AI risk model, which in turn performed 4–26% better than Tyrer–Cuzick v8, across multiple subgroups of women defined by breast cancer risk factors and by cancer subtypes.

The increase in discriminatory performance using the lifestyle/familial-expanded AI risk model over the imaging-only AI risk model could improve personalized screening and risk-reducing interventions. At the same time, an imaging-only risk algorithm provides several advantages over a model that requires data from additional sources. Image data are available for all women attending screening and an image-based algorithm provides consistent scores across a screening population. A lifestyle/familial-expanded AI risk model may add additional time and costs for implementation and is based on self-reporting prone to missing data and recall bias.

We estimated the model performance differences in subgroups using dichotomized risk factors to enable higher sample sizes and to decrease the likelihood of reporting false positive/negative results [36]. The lifestyle/familial-expanded AI risk model predicted risks non-significantly different compared to the imaging-only AI risk model in most of the investigated risk subgroups. This indicates that women with increased lifestyle/familial risk exposures also have increased exposures of the mammographic features in the imaging-only AI risk model. However, several subgroups of women defined by risk factors (alcohol intake, menopause status, age at menarche, height, BMI, age at first childbirth, and benign breast disease) were better identified using the lifestyle/familial-expanded AI risk model.

Compared with the imaging-only AI risk model, the discriminatory performance of the Tyrer–Cuzick v8 model was not significantly different in approximately three quarters of all investigated subgroups. In the remaining subgroups, the AUCs were significantly lower in the range of 4–26 percent with the largest AUC difference for stage 2 or later cancers.

In the U.S., approximately 12% of women of screening age have a lifetime risk of ≥20% and are eligible for supplemental screening with breast MRI [37]. In our study, we found that the PPV for risk prediction was significantly lower for Tyrer–Cuzick v8 than for lifestyle/familial-expanded image-derived AI risk model in 12% of women at the highest risk. We did not find any subgroup of women who had a higher discriminatory performance using the Tyrer–Cuzick v8 compared to the imaging-only model or the lifestyle/familial-expanded AI risk model. In addition, we did not find that women with a family history of breast cancer had significantly increased discriminatory performance using the lifestyle/familial-expanded AI risk model compared to the imaging-only AI risk model.

The lifestyle/familial-expanded AI risk model showed an advantage over the imaging-only AI risk model for identifying ER-positive breast cancers across the 1–10-year follow-up. Women therefore could have an advantage in having their risk assessed using the lifestyle/familial-expanded model when considering prophylactic endocrine therapy to reduce their risk of developing breast cancer [38].

Image-based models have been reported to perform well in the short term and are designed to personalize breast cancer screening [3,17,39]. Risk assessment in the short term identifies women who may benefit from supplemental screening or from a shorter screening interval to help identify breast cancers at an earlier stage. We did not see a significant advantage in identifying symptomatic and stage 2 or later cancers in the short term using the lifestyle/familial-expanded model over the imaging-only AI risk model.

Guidelines such as the NICE and USPSTF support the use of lifestyle/familial-based and genetic risk models for supplemental screening and risk-reducing intervention [14,15]. Considering that we observed a considerable and consistent discriminatory advantage using an image-derived AI-based risk model over a traditional lifestyle/familial-based risk model, investigations should evaluate extending guidelines to incorporate such newer, image-derived risk tools.

This study has several limitations. We investigated the model performances in a Swedish large-scale screening population for women at general risk, where the vast majority of women were white, attended biennial screening using digital mammography (GE, Philips, Sectra, Siemens, Fuji machines), and had a recall rate of ~3% with a cancer detection rate of ~5/1000 exams [40]. Our comparison with Tyrer–Cuzick v8 was limited by the fact that we did not have information on ovarian cancer in second-degree relatives and that we did not use a continuous measure for mammographic density. Further studies are needed investigating the generalizability of our reported results in independent screening programs. Further studies are also needed to investigate the extent of existing versus developing breast cancers in a ten-year time range after baseline. This may have consequences for strategies for improving breast cancer screening versus primary prevention. Studies are also needed to compare the lifestyle/familial-expanded AI risk model with the Tyrer–Cuzick v8 model and to investigate the addition of genetic determinants of breast cancer to the risk model.

## 5. Conclusions

A lifestyle/familial-expanded image-derived AI-based risk model showed a higher performance for capturing women who may benefit from risk reduction intervention and/or shorter screening intervals or supplemental screening compared to an imaging-only AI-based risk model and the clinical Tyrer–Cuzick v8 model. The higher performance was observed between 1 and 10 years after risk assessment across multiple subgroups of women defined by breast cancer risk factors and by cancer subtypes.

## Figures and Tables

**Figure 1 cancers-15-03246-f001:**
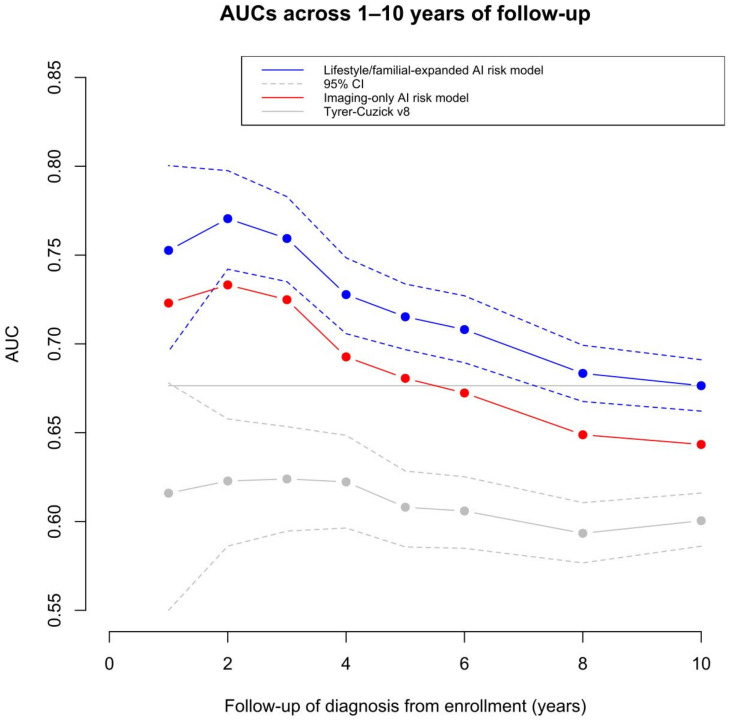
Age adjusted discriminatory performance of the imaging-only AI risk model, the lifestyle/familial-expanded AI risk model, and the Tyrer–Cuzick v8 risk model throughout the first years after enrollment. The 95% confidence intervals are presented for the lifestyle/familial-expanded AI risk model and Tyrer–Cuzick v8 model.

**Figure 2 cancers-15-03246-f002:**
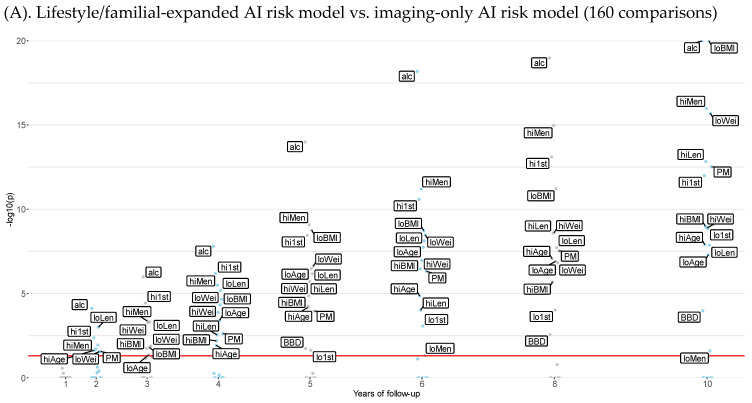
Manhattan plot of *p*-values denoting the significance of age adjusted AUC differences across 1–10 years of follow-up in subgroups of women by risk factor when comparing the lifestyle/familial-expanded AI risk model with the imaging-only AI risk model, which in turn is compared with Tyrer–Cuzick v8. The subgroups of women by risk factor are above/below median: age, length, weight, Body Mass Index (BMI); at least monthly intake of alcohol, current regular smoker; above/below median age at menarche, age at first childbirth, mammographic density; menopause, current use of hormone replacement therapy (HRT), benign breast disease, family history of breast cancer, 2nd/3rd degree relative with breast cancer, and ovarian cancer in the family. The red horizontal line represents the significance threshold (*p* = 0.05) after adjusting the *p*-values for multiple comparison using the Holm–Bonferroni method.

**Figure 3 cancers-15-03246-f003:**
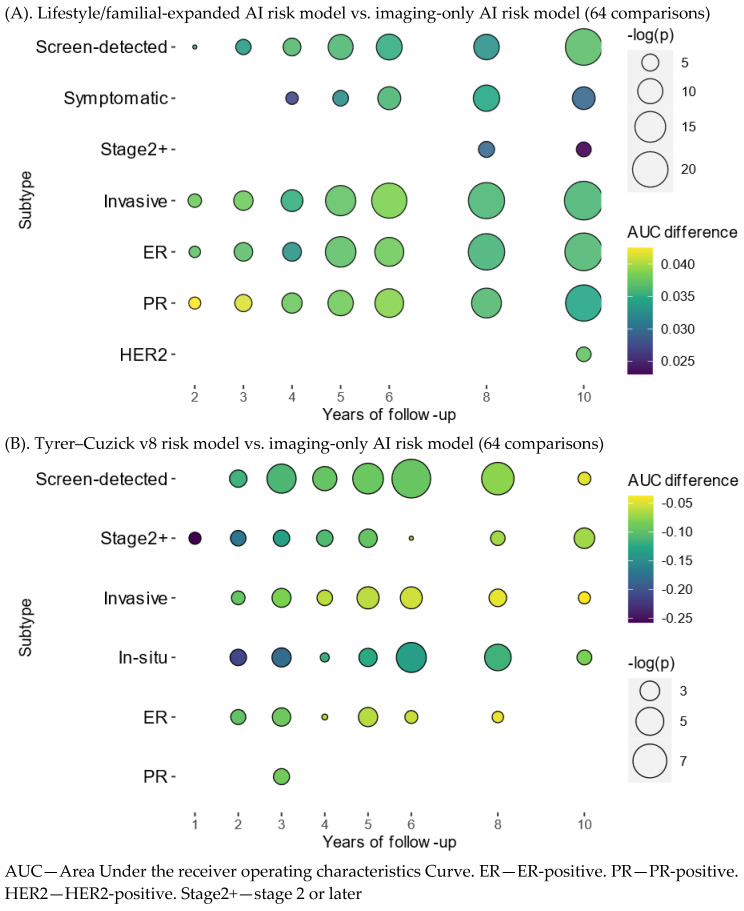
Bubble plot of *p*-values and AUC differences across 1–10 years of follow-up in subtypes of cancers comparing the lifestyle/familial-expanded AI risk model with the imaging-only AI risk model, which in turn is compared with Tyrer–Cuzick v8. The plot presents significantly different AUCs (*p* < 0.05) that were extracted after Holm–Bonferroni adjustment for multiple comparison. The bubble sizes represent minus-log-transformed *p*-values and the bubble colors represent age adjusted AUC point estimate differences. The investigated subtypes are mode of detection (screen-detected, symptomatic), stage (stage 1 or earlier, stage 2 or later), invasiveness (invasive, in situ), estrogen-receptor (ER), progesterone-receptor (PR), and human epidermal growth factor receptor 2 (HER2).

**Figure 4 cancers-15-03246-f004:**
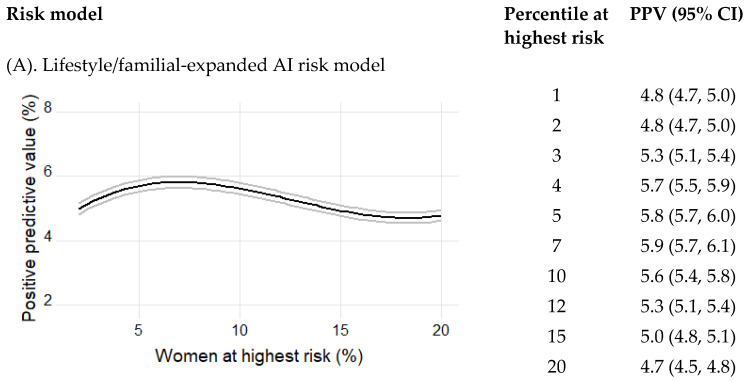
Positive predictive values of women at the highest risk of breast cancer after 10 years of follow-up. The plots present the positive predictive value (the proportion of breast cancers) captured by the model at study baseline among the 1–20% of women at the highest risk of breast cancer as predicted by the image-based risk model and the Tyrer–Cuzick v8 risk model, respectively.

**Table 1 cancers-15-03246-t001:** Study participant characteristics at enrollment including N = 9574 study participants in the case–cohort with up to 10 years of follow-up.

Characteristic	Subcohort ^1^, N = 8110 ^2^	Cases ^1^, N = 1661 ^2^	*p*-Value ^3^
Age	53.88 (9.86)	56.52 (9.53)	<0.01
Height (cm)	166.63 (6.01)	166.93 (5.88)	0.04
Weight (kg)	70.10 (12.54)	70.82 (12.27)	<0.01
BMI	25.24 (4.31)	25.38 (4.26)	0.03
At least monthly alcohol use last year	6461/8054 (80%)	1354/1650 (82%)	0.10
Regular smoking in last year	998/8093 (12%)	197/1658 (12%)	0.65
Age at menarche	13.07 (1.47)	13.11 (1.46)	0.31
Age at first childbirth	27.21 (5.29)	27.20 (5.21)	0.87
Postmenopausal	4411/8110 (54%)	1103/1661 (66%)	<0.01
Current use of HRT ^4^	304/7687 (4.0%)	106/1566 (6.8%)	<0.01
Benign breast disease	1756/7930 (22%)	517/1622 (32%)	<0.01
Breast cancer in 1st-degree family			<0.01
No family history	7069/8057 (88%)	1288/1643 (78%)	
Onset age < 50	253/8057 (3.1%)	90/1643 (5.5%)	
Onset age ≥ 50	735/8057 (9.1%)	265/1643 (16%)	
Breast cancer in 2nd/3rd-degree relative	730/5492 (13%)	168/1075 (16%)	0.03
Family history of ovarian cancer	308/7870 (3.9%)	69/1591 (4.3%)	0.36
Percent mammographic density	25.54 (19.45)	28.53 (19.43)	<0.01
Mammographic density above median	3989/8110 (49%)	923/1661 (56%)	<0.01
Imaging-only risk model, 2-year risk score	0.63 (0.95)	1.19 (1.85)	<0.01
Expanded AI risk model, 2-year risk score ^5^	0.66 (1.07)	1.37 (2.04)	<0.01
Tyrer–Cuzick v8, 2-year risk score	0.60 (0.39)	0.79 (0.53)	<0.01
Tyrer–Cuzick v8, 10-year risk score	3.09 (1.86)	3.96 (2.49)	<0.01

^1^ Subcohort of non-cases and the 197 incident breast cancers diagnosed in the subcohort. For cases, all incident breast cases outside and within the subcohort are included. ^2^ Mean (SD); n/N (%). ^3^ Welch Two Sample *t*-test; Pearson’s Chi-squared test; Fisher’s exact test; Wilcoxon rank sum test. ^4^ Hormone replacement therapy. ^5^ Lifestyle/familial-expanded AI risk model.

## Data Availability

The datasets for this study fall under GDPR legislation and are available upon reasonable request.

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
