# Peer review of "A Clinical Risk Model for Personalized Screening and Prevention of Breast Cancer"

_cancers, 2023, doi:10.3390/cancers15123246_

Round 1

Reviewer 1 Report

A Clinical Risk Model for Personalized Screening and Prevention of Breast Cancer

The aim of this study is to evaluate the long-term predictive performance using an image derived AI-based risk model expanded with lifestyle risk factors and family history of breast cancer, comparing it with the only-AI-based risk model and the IBIS model.

The authors use data from the KARMA study, 1,661 cases, and 8,110 controls for this study.

This main topic of the article is undoubtedly very relevant due to the consistent evidence about how risk-based personalized screening can improve the risk-benefit balance of this procedure, and the need in the last years to improve breast cancer prediction models to achieve this goal. The AI image-based technology seems that can really help in the discrimination of women by their personal risk.

Overall, the methodology used seems to be sound and the article is well-written. This is an article most probably worth publishing, as it is an interesting and very timely study, however, I have some comments that are feasible to be addressed by the authors and can contribute, in my opinion, to improve the paper.

1. What strikes me most in the article is that the authors do not explain how they develop the extended AI Risk model at all. I know the profound AI risk and the previous work of this group, but I read this paper and I feel there is some kind of “black box” in the methodology, that I think it can be easily fixed with a brief paragraph on methodology. There are many different ways and many different statistical methodologies to deal with adding new variables to a preexisting model (or to develop a new model) and how they affect absolute risk (using generalized models both fixed, mixed, survival models, machine learning models, etc...). I can't find anywhere how the authors do it. 

For example: is the risk of the AI risk model being updated with the relative risks of the other variables or the AI extended model is created independently of the previous one? If it’s the first option, I would like to know how the relative risks were estimated and if possible, if interactions were considered, such as an interaction between the only-AI model risk and the relative risk conferred by the variable "previous benign breast disease disorders” which may (or not) be correlated.

2. I understand from the article that the validation statistics (in this case discrimination) used in the extended AI risk model were calculated on the same population with which the model estimators were calculated. This is not at all usual and usually causes overestimation of these estimates. The usual procedure is to use different subcohorts for the estimation of the model and for the validation of the same, even if come from the same cohort (internal validation). Did the authors consider split validation or cross-validation?

3. There is something I don't understand about the study population (sorry about that, I'm not that used to case controls on this topic). I understand that, as we do with the longitudinal screening cohorts when developing predictive models, the mammogram at cancer diagnosis is never used, since the previous ones have to be used. For example, if you find a breast cancer diagnosis in screening mammography you will use information from, at least, 2 years before this mammogram. In this case, which mammograms are used? Because as a case-control study, I understand that you are only using one mammogram per woman since it is not a longitudinal study, but a cross-sectional one. 

If you have 10 mammograms in a control, which one do you use, the first?

And if you have a control with 10 mammograms, which one do you use again, also the first one?

This seems to me to be an important point for two reasons: 1) it should be better reflected in the article. 2) If you take the first one, why is there a significant difference in age (table 1)? I understand that if you take the first one from both cases and controls, and in cases that are previous to the cancer diagnosis, the mean age in both should be the same, otherwise, it would imply a possible selection bias since the age itself is contributing risk to that woman.

4. Did the authors consider adding the relative risks of the risk factors? I think it might be interesting to add it to a table, maybe in the supplemental material.  There is a lot of information in the supplementary tables that I do not see as important, but I understand that the reader of an article like this might find it interesting to see the contribution of each of these variables, not only to the AUC but also to the risk of each woman. How much does drinking a lot of alcohol affect her individual risk? etc... 

5. The authors analyze the discriminative power of the model and not the calibration power. This can make sense, when talking about personalization in screening, the main aim is that the model discriminates. But then in Table 2, they stratify the population into the different groups of the NICE guidelines using the cut-off points for absolute risk. I believe that this should not be done without first checking, by means of any of the many calibration statistics, that the model is correctly calibrated because no matter how well the model discriminates, discrimination and calibration are not correlated. In other words, if the model discriminates well but calibrates poorly, it will not be putting women in the risk group they are in, and this would not be positive either.

6. In statistical analysis authors say: “To estimate the risk performance for a certain year of follow-up, we used the subcohort and the incident breast cancers that developed after study entry, between 3 months and that year”. I have some questions about this sentence:

a. I think risk performance is misleading. If you are talking about how you estimate AUC, consider changing it for “discrimination performance” (or something regarding discrimination” as it is not the same, as I comment in the previous point I have found models with really high AUCs but poorly calibrated.

b. When you say the “subcohort” I understand that is the subcohort of women follow-up until that point right? So, if you want to estimate the 2-year AUC you pick all your women, and for example for the 6 years AUC only those women with a follow-up time of 6 years or higher, right?

c. I understand that there is not any possibility of, after having a recall for a mammogram, having a cancer diagnosis derived from that mammogram in a period higher than 3 months?

7. Regarding interval cancers, which authors call symptomatic. What is the reason for the high number of them? 40% is a really high amount, at least in our context they should always be less than 20%, as it’s a number that is directly related to the sensitivity of screening. Just curiosity.

8. From Supplementary table B2 strikes me that Use of HRT has 0.51 AUC in year 1 and jumps to 0.70 in year 2. Do the authors have any idea of why? Even more, this variable is the only one that has a higher AUC in the image only than in the extended ai risk model. Why is that?

9. How did the authors estimate the IBIS model estimates risk across the 1-10-year period? Usually, the IBIS model only gives information on the 5-year risk and the 10-year risk, as it is not a model that was created to perform short-term estimations.

10. Page 3: “The Tyrer-Cuzick v8 risk model was used as a clinical comparison tool [21]. Tyrer-Cuzick provides breast cancer risk based on lifestyle and familial risk factors, i.e. age, height, weight, age at menarche, age at first childbirth, menopausal status, use of HRT, previous benign breast disorders, first and second-degree family history of breast cancer, first-degree family history of ovarian breast cancer, and mammographic density categorized into BI-RADS categories”. The Tyrer-Cuzick model has more variables included. Did the authors use all the information available of the women for the Tyrer Cuzick information or only the ones available in the extended AI risk model? 

11. In table 1 I am surprised by the p-value of age at menarche and age at first childbirth, as are both variables that I’ve usually found significant in terms of the proportion of cancer at least in my models/cohorts. Another variable that seems not to be significant is regular smoking in the last year. Were the relative risks of these variables significant? Why regular smoking is included in the lifestyle/familial-expanded AI-risk model if no significance, and why is the decision to not include age at menarche or age at first birth? As I think age at menarche and age at first birth has already been proven to affect risk of breast cancer, if you do not add them to the model because of the lack of significance consider adding a sentence in limitations (maybe is for sample size?).

12. I understand that profound AI risk gives both the percent of density and the BI-Rads density category, right?

13. Is data available? There are precedents for researchers releasing such data used to fit risk models. For example, you can access a modified version of the BCSC data used for their model, where categories have been coded (e.g. not individual year of age). I'd encourage the authors to consider trying to do this if at all possible. It would also be worth making your code available, for transparency of the statistical methods used.

Author Response

Reviewer 1

A Clinical Risk Model for Personalized Screening and Prevention of Breast Cancer

The aim of this study is to evaluate the long-term predictive performance using an image derived AI-based risk model expanded with lifestyle risk factors and family history of breast cancer, comparing it with the only-AI-based risk model and the IBIS model.

The authors use data from the KARMA study, 1,661 cases, and 8,110 controls for this study.

This main topic of the article is undoubtedly very relevant due to the consistent evidence about how risk-based personalized screening can improve the risk-benefit balance of this procedure, and the need in the last years to improve breast cancer prediction models to achieve this goal. The AI image-based technology seems that can really help in the discrimination of women by their personal risk.

Overall, the methodology used seems to be sound and the article is well-written. This is an article most probably worth publishing, as it is an interesting and very timely study, however, I have some comments that are feasible to be addressed by the authors and can contribute, in my opinion, to improve the paper.

  1. What strikes me most in the article is that the authors do not explain how they develop the extended AI Risk model at all. I know the profound AI risk and the previous work of this group, but I read this paper and I feel there is some kind of “black box” in the methodology, that I think it can be easily fixed with a brief paragraph on methodology. There are many different ways and many different statistical methodologies to deal with adding new variables to a preexisting model (or to develop a new model) and how they affect absolute risk (using generalized models both fixed, mixed, survival models, machine learning models, etc...). I can't find anywhere how the authors do it. 

For example: is the risk of the AI risk model being updated with the relative risks of the other variables or the AI extended model is created independently of the previous one? If it’s the first option, I would like to know how the relative risks were estimated and if possible, if interactions were considered, such as an interaction between the only-AI model risk and the relative risk conferred by the variable "previous benign breast disease disorders” which may (or not) be correlated.

R1-1. Thank you for very relevant questions that need addressing in the manuscript. In this manuscript, we used the previously developed model described in the Radiology paper from 2020 (DOI: 10.1148/radiol.2020201620). We compare Model 1 (imaging-only) with Model 2 (expanded with lifestyle/familial risk factors). We have now included the following description in the methods section 2.2:

“The lifestyle/familial-expanded AI-risk model was constructed using the mammographic features in the imaging-only risk model in combination with the lifestyle/familial factors as previously described (Model 2) [17]”.

We are happy to repeat the description of the model construct as part of this paper. If requested, we will contact Radiology and ask for reprinting this. For Model 2 we re-estimate the estimates for the mammographic features as additional lifestyle/familial factors are included. We also account for menopausal status interaction by providing different set of estimates for pre- and postmenopausal women. We did not see an added value using benign breast disease in the model.

  1. I understand from the article that the validation statistics (in this case discrimination) used in the extended AI risk model were calculated on the same population with which the model estimators were calculated. This is not at all usual and usually causes overestimation of these estimates. The usual procedure is to use different subcohorts for the estimation of the model and for the validation of the same, even if come from the same cohort (internal validation). Did the authors consider split validation or cross-validation?

R1-2. Thank you for this very important question. We want to clarify that the previous development of the model was performed on the same underlying Swedish screening cohort but using a smaller sample size. In our current manuscript we used an independent set of mammograms from study baseline. With the development of breast cancers over the years, we also had a longer follow-up (up to 10 years) compared to 2 years previously. We want to clarify this limitation of the manuscript in the limitations section where we clearly want to point out that further studies are needed to investigate our reported results:

“Further studies are needed investigating the generalizability of our reported results in independent screening programs”.

  1. There is something I don't understand about the study population (sorry about that, I'm not that used to case controls on this topic). I understand that, as we do with the longitudinal screening cohorts when developing predictive models, the mammogram at cancer diagnosis is never used, since the previous ones have to be used. For example, if you find a breast cancer diagnosis in screening mammography you will use information from, at least, 2 years before this mammogram. In this case, which mammograms are used? Because as a case-control study, I understand that you are only using one mammogram per woman since it is not a longitudinal study, but a cross-sectional one. 

If you have 10 mammograms in a control, which one do you use, the first?

And if you have a control with 10 mammograms, which one do you use again, also the first one?

This seems to me to be an important point for two reasons: 1) it should be better reflected in the article. 2) If you take the first one, why is there a significant difference in age (table 1)? I understand that if you take the first one from both cases and controls, and in cases that are previous to the cancer diagnosis, the mean age in both should be the same, otherwise, it would imply a possible selection bias since the age itself is contributing risk to that woman.

R1-3. We fully agree that this is a crucial question. We clarify that in our study we used a case cohort study design and used the mammograms at study entry, i.e. when women entered the study between 2010-2013, to assess the risk. We then follow the women till 2022 for breast cancer outcome. We have clarified this in the methods section 2.2 as follows:

“Absolute risks of breast cancer were calculated for the three models based on their inclusion of risk factors, risk factor prevalence, and Swedish national statistics on breast cancer incidence rate and competing mortality risk [18]. Risk assessment was performed at study enrollment. Breast cancer status and mode of detection were retrieved for breast cancers diagnosed up to January 2022 from the National Quality register for Breast Cancer (NKBC) register through linkage using unique the Swe-dish personal identification numbers”.

We also clarify that we used the KARMA mammography screening population which is the largest epidemiological screening cohort for breast cancer in Sweden (DOI: 10.1093/ije/dyw357). As we recruited a screening cohort and the mean age of women attending screening and the mean age when women develop breast cancer naturally differs, we see an age difference in Table 1. As is well noted, it is key in the analysis to adjust for age differences between cases and controls. We therefore adjust all our analyses for the age difference. We describe it as follows in the methods section 2.3:

“Area Under the receiver operating characteristics Curve (AUC) estimated the discriminatory performance of the models across the 1-10 year follow-up period after age adjustment [26,27]”.

In this way, our analyses were not biased by age.

  1. Did the authors consider adding the relative risks of the risk factors? I think it might be interesting to add it to a table, maybe in the supplemental material.  There is a lot of information in the supplementary tables that I do not see as important, but I understand that the reader of an article like this might find it interesting to see the contribution of each of these variables, not only to the AUC but also to the risk of each woman. How much does drinking a lot of alcohol affect her individual risk? etc... 

R1-4. Thank you for pointing this out. We fully agree that this information is helpful. For our current manuscript, we do not re-estimate relative risks. In our original Radiology paper in 2020 we described in detail the relative risks that the model uses (Supplementary Table 1). If requested, we are happy to contact Radiology for potentially repeating the relative risk information also in this paper for easier reading. We found in the Radiology 2020 paper that alcohol in premenopausal women had a relative risk of 1.18 and 1.14 in postmenopausal women for Model 2. This was based on an alcohol drinking variable that was dichotomized into drinking alcohol at least monthly or less/never. In future work we will re-estimate the relative risks where mammographic features also will have time-dependent estimates.

  1. The authors analyze the discriminative power of the model and not the calibration power. This can make sense, when talking about personalization in screening, the main aim is that the model discriminates. But then in Table 2, they stratify the population into the different groups of the NICE guidelines using the cut-off points for absolute risk. I believe that this should not be done without first checking, by means of any of the many calibration statistics, that the model is correctly calibrated because no matter how well the model discriminates, discrimination and calibration are not correlated. In other words, if the model discriminates well but calibrates poorly, it will not be putting women in the risk group they are in, and this would not be positive either.

R1-5. We fully agree with the reviewer, and we have now put forward another metrics of positive predictive value (PPV), that we had as supplementary material, that assesses the PPV in women at the highest risk. We assess the PPV in the top 1-20% of women at highest risk and compare the three models. We also suggest making this part of the main result in the manuscript for a better comparison between the models. This is now Figure 4 in the manuscript. We also updated in results and in the abstract as follows:

“Among the 5% of women at highest risk, the PPV was 5.8% using the lifestyle/familial-expanded model compared with 5.3% using the imaging-only model, p<0.01, and 4.6% for Tyrer-Cuzick, p<0.01.”

  1. In statistical analysis authors say: “To estimate the risk performance for a certain year of follow-up, we used the subcohort and the incident breast cancers that developed after study entry, between 3 months and that year”. I have some questions about this sentence:
  2. I think risk performance is misleading. If you are talking about how you estimate AUC, consider changing it for “discrimination performance” (or something regarding discrimination” as it is not the same, as I comment in the previous point I have found models with really high AUCs but poorly calibrated.
  3. When you say the “subcohort” I understand that is the subcohort of women follow-up until that point right? So, if you want to estimate the 2-year AUC you pick all your women, and for example for the 6 years AUC only those women with a follow-up time of 6 years or higher, right?
  4. I understand that there is not any possibility of, after having a recall for a mammogram, having a cancer diagnosis derived from that mammogram in a period higher than 3 months?

 R1-6. Thank you for pointing this out. We have changed to “discrimination performance” as requested. We confirm that it is correct that we follow the subgroup of women (cases and controls) up to each year of follow-up in the AUC estimations. We also confirm that the lead time of 3 months is there for excluding the possibility that there was a late diagnosis following the prior mammogram. We confirm that we did not have any diagnosis of breast cancer following the prior mammogram after 3 months. The vast majority of breast cancers were diagnosed within 1 month.

  1. Regarding interval cancers, which authors call symptomatic. What is the reason for the high number of them? 40% is a really high amount, at least in our context they should always be less than 20%, as it’s a number that is directly related to the sensitivity of screening. Just curiosity.

R1-7. We see a high proportion of symptomatic cancers in our cohort. The same is true for the entire Swedish mammography screening program. In a recent report from the Swedish National Board of Health and Well-fare it was stated that 35% of all cancers are interval cancers in Sweden. Unfortunately, the reference is in Swedish: Socialstyrelsen, Nationell utvärdering – bröstcancerscreening med mammografi. Socialstyrelsen Artikelnummer 2022-6-7958, 2022. In Sweden, women are screened from age 40 in an age where mammographic density is higher and lower screening sensitivity to a higher degree compared with women in later ages.

  1. From Supplementary table B2 strikes me that Use of HRT has 0.51 AUC in year 1 and jumps to 0.70 in year 2. Do the authors have any idea of why? Even more, this variable is the only one that has a higher AUC in the image only than in the extended ai risk model. Why is that?

R1-8. We believe that given the few events of cancers occurring in 1 year related to HRT, this may be a chance finding and we could not find a significant difference in Figure 2 between the two models for use of HRT in these years.

  1. How did the authors estimate the IBIS model estimates risk across the 1-10-year period? Usually, the IBIS model only gives information on the 5-year risk and the 10-year risk, as it is not a model that was created to perform short-term estimations.

R1-9. We have access to a more detailed version of IBIS from the authors of IBIS, which makes it possible for us to assess risk at several risk projection times. We presented results in Table 2 using 10-year risk projection time. In Supplementary Table B5 we used 5-year risk projection time. We also internally tested 1-10 year risk projection times. This did not change the conclusion or AUC estimates other than marginally. We fully agree that the IBIS model is not designed for generating short-term estimations. We believe that image-based models are better suited for this task and could be a better alternative for identifying women who need supplemental screening in the U.S.

  1. Page 3: “The Tyrer-Cuzick v8 risk model was used as a clinical comparison tool [21]. Tyrer-Cuzick provides breast cancer risk based on lifestyle and familial risk factors, i.e. age, height, weight, age at menarche, age at first childbirth, menopausal status, use of HRT, previous benign breast disorders, first and second-degree family history of breast cancer, first-degree family history of ovarian breast cancer, and mammographic density categorized into BI-RADS categories”. The Tyrer-Cuzick model has more variables included. Did the authors use all the information available of the women for the Tyrer Cuzick information or only the ones available in the extended AI risk model? 

R1-10. Thank you for pointing this out. We have now included an updated description in the methods section 2.2 as follows:

“We did not have information regarding history of ovarian cancers in second degree relatives”.

We also included in the limitations section:

“Our comparison with Tyrer-Cuzick was limited by the fact that we did not have information on ovarian cancer in second degree relatives”.

  1. In table 1 I am surprised by the p-value of age at menarche and age at first childbirth, as are both variables that I’ve usually found significant in terms of the proportion of cancer at least in my models/cohorts. Another variable that seems not to be significant is regular smoking in the last year. Were the relative risks of these variables significant? Why regular smoking is included in the lifestyle/familial-expanded AI-risk model if no significance, and why is the decision to not include age at menarche or age at first birth? As I think age at menarche and age at first birth has already been proven to affect risk of breast cancer, if you do not add them to the model because of the lack of significance consider adding a sentence in limitations (maybe is for sample size?).

R1-11. Thank you for these interesting points on risk factors. Our Model 2 was not developed in this study cohort, but as part of the previous Radiology 2020 paper. Our approach was to start by using the mammographic features, our Model 1. We then investigated what additional lifestyle/familial risk factors could add to the image-based model. In this analysis, we did not find age at menarche and age at first childbirth to be additive to the model. From analyses related to other projects, we know that e.g. hormonal related factors are captured in the level of mammographic density a women has later in her life. We believe that this is the reason why we didn’t see an additional value of including age at menarche and age at first childbirth. We see significant associations with these factors, how they are not additive to our base model. However, regular smoking showed an additive predictive value.

  1. I understand that profound AI risk gives both the percent of density and the BI-Rads density category, right?

R1-12. Yes, it is correct that it provides both classifications.

  1. Is data available? There are precedents for researchers releasing such data used to fit risk models. For example, you can access a modified version of the BCSC data used for their model, where categories have been coded (e.g. not individual year of age). I'd encourage the authors to consider trying to do this if at all possible. It would also be worth making your code available, for transparency of the statistical methods used.

R1-13. We generally share data from the Karma Cohort and provide data upon reasonable request directed to us as authors of papers. There is a Data Access Committee and instructions on how to get access to data at our website [karmastudy.org]. In our current manuscript, we did not develop the model, but we evaluated our existing model for its discriminatory performance in a longer-term follow-up. We also like to clarify that we fully comply with the transparency and Cancers policy.

Reviewer 2 Report

interesting manoscript,

actually for better understanding, it should be stated tat all women were  observed for at least 10 years, otherwise OC analysis might be incoorect.

Author Response

Reviewer 2

interesting manoscript,

actually for better understanding, it should be stated tat all women were  observed for at least 10 years, otherwise OC analysis might be incoorect.

R2-1. Thank you for the comment. We clarify that the follow-up time for all women in the study was at least 10 years. We have now updated the results section 3.2 as follows:

“The follow-up period for the women in this study was over 10 years and, the time from the mammogram at enrollment to breast cancer diagnosis ranged from 3 months to over 10 years, Supplementary Figure A1”.

Reviewer 3 Report

The authors have presented a manuscript that examined a clinical risk model for personalized screening and prevention of breast cancer. The  lifestyle/familial-expanded AI-model was compared to the imaging-only AI-risk model and the Tyrer-Cuzick and Gail models. The lifestyle/familial-expanded AI-risk model showed higher performance for both long-term and short-term risk assessment compared with imaging-only and Tyrer-Cuzick models. 

The manuscript is well written and interesting to the reader. 

Abstract: This is concise. However, the author should clearly state the study study objective here.

Introduction: Nicely written.

Methods: Under Risk Factors at Enrollment and Breast Cancers at Follow-up, the survey instrument should be briefly described, and the reliability and validity indices mentioned, for the The lifestyle/familial-expanded AI-risk model that used BMI, alcohol, smoking, use of hormone replacement therapy (HRT), and first degree family history of breast cancer.

Results: These were well presented. The tables and figures are nicely done and very informative.

Discussion: This was nicely done.

Best of luck!

Author Response

Reviewer 3

The authors have presented a manuscript that examined a clinical risk model for personalized screening and prevention of breast cancer. The  lifestyle/familial-expanded AI-model was compared to the imaging-only AI-risk model and the Tyrer-Cuzick and Gail models. The lifestyle/familial-expanded AI-risk model showed higher performance for both long-term and short-term risk assessment compared with imaging-only and Tyrer-Cuzick models. 

The manuscript is well written and interesting to the reader. 

Abstract: This is concise. However, the author should clearly state the study study objective here.

R3-1.Thank you for the comments. We have now clarified as following:

“Image-derived artificial intelligence (AI) risk models have shown promise in identifying high-risk women in the short-term. The long-term performance of image-derived risk models expanded with clinical factors has not been investigated”.

Introduction: Nicely written.

Methods: Under Risk Factors at Enrollment and Breast Cancers at Follow-up, the survey instrument should be briefly described, and the reliability and validity indices mentioned, for the The lifestyle/familial-expanded AI-risk model that used BMI, alcohol, smoking, use of hormone replacement therapy (HRT), and first degree family history of breast cancer.

R3-2. We have now included the reference for the questionnaire we used for collecting the risk factors for this study as follows:

“The factors were extracted from the questionnaire developed as part of the KARMA project [16]”.

Results: These were well presented. The tables and figures are nicely done and very informative.

Discussion: This was nicely done.

Best of luck!

Reviewer 4 Report

In a manuscript entitled "A Clinical Risk Model for Personalized Screening and Prevention of Breast Cancer" the authors elaborated an Image-derived artificial intelligence risk models from the case-cohort study of women selected from the mammography screening cohort. I have several concerns that need to address for the improvement of the manuscript.

1. Pleas highlighted the limitations at the review level in the manuscript.

2. The authors should re-read the full manuscript and reorganize chronologically the abbreviations and correct typos. 

3. The references need reorganizing according to Cancers journal references style. 

4. Please report the supplementary Figures and Tables as a supplementary materials file. 

5. In the manuscript, the following references may be considered:

DOI: 10.1007/s10549-021-06445-8

DOI: 10.1016/j.lfs.2021.119110

6. It is better if the authors include more specific points of the findings at Conclusion section to reflect the overall understanding of the work.

Author Response

Reviewer 4

In a manuscript entitled "A Clinical Risk Model for Personalized Screening and Prevention of Breast Cancer" the authors elaborated an Image-derived artificial intelligence risk models from the case-cohort study of women selected from the mammography screening cohort. I have several concerns that need to address for the improvement of the manuscript.

  1. Pleas highlighted the limitations at the review level in the manuscript.

R4-1. Thank you for pointing this out. We now identified needs for improvement and highlighted the limitation with the Tyrer-Cuzick model in the limitations section as follows:

“Our comparison with Tyrer-Cuzick was limited by the fact that we did not have information on breast cancers in third degree relatives and not on second degree ovarian cancers”.

  1. The authors should re-read the full manuscript and reorganize chronologically the abbreviations and correct typos. 

R4-2. We have now re-read the full manuscript and reorganized the abbreviations chronologically. In addition, we corrected typos.

  1. The references need reorganizing according to Cancers journal references style.

R4-3. We have updated to Cancers journal references style.

  1. Please report the supplementary Figures and Tables as a supplementary materials file.

R4-4. We have now done this.

  1. In the manuscript, the following references may be considered:

DOI: 10.1007/s10549-021-06445-8

DOI: 10.1016/j.lfs.2021.119110

R4-5. Thank you for the suggestions. We have added the first reference.

  1. It is better if the authors include more specific points of the findings at Conclusion section to reflect the overall understanding of the work.

R4-6. Thank you for pointing this out. We have now added in the conclusion the following:

“The higher performance was observed between 1-10 years after risk assessment across multiple subgroups of women defined by breast cancer risk factors and by cancer subtypes”.

Reviewer 5 Report

The authors present interesting data on their AI model on mammography and show that this is enhanced by using in an extended model with other risk factors. It is clear from the drop off in AUC after 2 years that the main advantage of the AI model is identification of cancers that are already present. It is absolutely not a valid comparison to assess against a clinical Tyrer-Cuzick model alone. If anything they should compare with their in house clinical model alone. For this to be useful internationally the authors need to show that their clinical model is at least equivalent or better than Tyrer-Cuzick without a density/AI input and with both models with an AI density input. It is not clear how any centre can gain access to the models to validate them

Specific points

1.        ‘In a case-cohort study, 8,110 women were randomly selected from women in age 40-74 participating in a Swedish mammography screening cohort’ -It seems unlikely the cases were randomly selected ?

2.       ‘The increased performances were observed in multiple risk subgroups and cancer subtypes. In the high-risk women per NICE guidelines, 22%, 19%, 7.2% of BCs were identified using the lifestyle/familial-expanded model, the imaging-only model, p<0.01, and Tyrer-Cuzick, p<0.01.’ -it is not appropriate to compare a risk model (Tyrer-Cuzick) that contains a density input with a mammography imaging model without at least imputing a basic mammographic density input such as BiRads. The only correct comparison is with Tyrer-Cuzick and the new lifestyle/familial component without AI

3.       ‘TyrerCuzick provides breast cancer risk based on lifestyle and familial risk factors, i.e. age, height, weight, age at menarche, age at first childbirth, menopausal status, use of HRT, previous benign breast disorders, first and second-degree family history of breast cancer, first degree family history of ovarian breast cancer, and mammographic density categorized into BI-RADS categories [20].’ -This is incorrect. Breast cancer family history in cousins (third degree relatives) is also included. Ovarian cancer is also included for second degree relatives. Mammographic density is also possible using continuous measures such as visual assessment and Volpara. Please correct this. Please also stipulate whether BiRads was entered in the T-C model

4.       ‘1The NICE guidelines 10-year absolute risk categories were general, moderate, and high using absolute risk cut-offs 3% and 8%.’ -its is NOT appropriate to use the 3% threshold for moderate risk except in women aged 40 years. A risk threshold of 5% has been accepted for women of screening age in the UK. It is clear from the % at high risk that BiRads has not been inputted into T-C

5.       ‘Compared with the imaging-only AI-risk model, the discriminatory performance of the  Tyrer-Cuzick v8 model was non-significantly different in approximately three quarters of all investigated subgroups. In the remaining subgroups, the AUCs were significantly lower in the range of 4-26 percent with the largest AUC difference for stage 2 or later cancers.’ -This is unsurprising as the AI model is detecting cancers that are already there but missed by the screening. As such these will be likely to be stage 2 at presentation. This is further evidence for the inappropriateness of assessing versus TC alone without a density input.

6.       The analyses should be completely redone either without the Tyrer-Cuzick comparison or with Tyrer-Cuzick PLUS a density input. The only valid comparison with the clinical only Tyrer-Cuzick would be the in house lifestyle and familial risk factors model WITHOUT AI.

7.       Why have the authors not used TC along with their AI density model as a comparison?

Please present data on the models from 2-10 years excluding the next mammogram and interval cancers to 2 years

Author Response

Reviewer 5

The authors present interesting data on their AI model on mammography and show that this is enhanced by using in an extended model with other risk factors. It is clear from the drop off in AUC after 2 years that the main advantage of the AI model is identification of cancers that are already present. It is absolutely not a valid comparison to assess against a clinical Tyrer-Cuzick model alone. If anything they should compare with their in house clinical model alone. For this to be useful internationally the authors need to show that their clinical model is at least equivalent or better than Tyrer-Cuzick without a density/AI input and with both models with an AI density input. It is not clear how any centre can gain access to the models to validate them

R5-1. Thank you for the comments. We want to clarify that we use Tyrer-Cuzick model v8 in this comparison and that it includes mammographic density. We do not compare the AI model with Tyrer-Cuzick alone. The purpose of our manuscript was to compare the AI models as constructed with Tyrer-Cuzick v8 as constructed. In our manuscript, we did not intend to create a new AI-based model without AI and we did not intend to develop Tyrer-Cuzick with AI. We do not see the purpose of making comparisons between non-existing models, where the AI-based model does not exist without the AI component and Tyrer-Cuzick does not exist with AI and how this is related to being internationally useful. We also like to clarify that the AI risk model is currently offered to clinics in U.S. and Europe.

Specific points

  1. ‘In a case-cohort study, 8,110 women were randomly selected from women in age 40-74 participating in a Swedish mammography screening cohort’ -It seems unlikely the cases were randomly selected ?

R5-2. Thank you for pointing this out. We have corrected the abstract by removing “randomly”.

  1. ‘The increased performances were observed in multiple risk subgroups and cancer subtypes. In the high-risk women per NICE guidelines, 22%, 19%, 7.2% of BCs were identified using the lifestyle/familial-expanded model, the imaging-only model, p<0.01, and Tyrer-Cuzick, p<0.01.’ -it is not appropriate to compare a risk model (Tyrer-Cuzick) that contains a density input with a mammography imaging model without at least imputing a basic mammographic density input such as BiRads. The only correct comparison is with Tyrer-Cuzick and the new lifestyle/familial component without AI

R5-3. We would like to clarify that Tyrer-Cuzick v8 includes mammographic density. In the introduction we have written that

“We studied the predictive performances up to 10-years after study-entry to identify women who may benefit from risk reducing intervention using the lifestyle/familial-expanded model compared to the imaging-only AI risk model, which in turn was compared to the clinically used Tyrer-Cuzick v8 risk model which includes mammographic density”.

We have also written in the methods that:

“The Tyrer-Cuzick v8 risk model was used as a clinical comparison tool” and that “Tyrer-Cuzick provides breast cancer risk based on lifestyle and familial risk factors, i.e. age, height, weight, age at menarche, age at first child-birth, menopausal status, use of HRT, previous benign breast disorders, first, second, and third-degree family history of breast cancer, first degree family history of ovarian breast cancer, and mammographic density categorized into BI-RADS categories”.

The purpose of our manuscript is to compare our current two models (Model 1using images only and Model 2 expanded with lifestyle/familial factors) with the reference model Tyrer-Cuzick v8. Our model is constructed using AI as a core component. We disagree on changing the purpose of the manuscript and constructing a new model including only the lifestyle/familial component for comparison with Tyrer-Cuzick (without density).

  1. ‘TyrerCuzick provides breast cancer risk based on lifestyle and familial risk factors, i.e. age, height, weight, age at menarche, age at first childbirth, menopausal status, use of HRT, previous benign breast disorders, first and second-degree family history of breast cancer, first degree family history of ovarian breast cancer, and mammographic density categorized into BI-RADS categories [20].’ -This is incorrect. Breast cancer family history in cousins (third degree relatives) is also included. Ovarian cancer is also included for second degree relatives. Mammographic density is also possible using continuous measures such as visual assessment and Volpara. Please correct this. Please also stipulate whether BiRads was entered in the T-C model

R5-4. Thank you for pointing this out. We have now expanded section 2.2. in Methods and included:

“We did not have information regarding history of ovarian cancer in second degree relatives”.

We also included in the limitation section the following:

“Our comparison with Tyrer-Cuzick was limited by the fact that we did not have information on ovarian cancers in second degree”.

We also clarify that we include BiRads density in the T-C model.

  1. ‘1The NICE guidelines 10-year absolute risk categories were general, moderate, and high using absolute risk cut-offs 3% and 8%.’ -its is NOT appropriate to use the 3% threshold for moderate risk except in women aged 40 years. A risk threshold of 5% has been accepted for women of screening age in the UK. It is clear from the % at high risk that BiRads has not been inputted into T-C

R5-5. We have now included the 5% cut-off for moderate risk for women in screening age as requested per the NICE guidelines. This is reported in supplementary table B5. We also clarify that Tyrer-Cuzick v8 has been used.

  1. ‘Compared with the imaging-only AI-risk model, the discriminatory performance of the  Tyrer-Cuzick v8 model was non-significantly different in approximately three quarters of all investigated subgroups. In the remaining subgroups, the AUCs were significantly lower in the range of 4-26 percent with the largest AUC difference for stage 2 or later cancers.’ -This is unsurprising as the AI model is detecting cancers that are already there but missed by the screening. As such these will be likely to be stage 2 at presentation. This is further evidence for the inappropriateness of assessing versus TC alone without a density input.

R5-6. We would like to point out that the results for stage 2 or later cancers were observed in the time range from 1-10 years of follow-up, Figure 3B. We observed a decreasing AUC difference in the range of 25-10% over time. We believe that this information is highly relevant to report on for increasing the understanding of existing versus developing breast cancers in a time range up to ten years after baseline. This may have consequences for strategies for improving breast cancer screening versus primary prevention. We also point out that Tyrer-Cuzick v8 in our comparison includes mammographic density.

Finally, it should be underlined that any risk model, except lifetime models, are likely to identify cancers that “are already” there. The sub clinical phase of most breast cancers are probably longer than 5 years.

  1. The analyses should be completely redone either without the Tyrer-Cuzick comparison or with Tyrer-Cuzick PLUS a density input. The only valid comparison with the clinical only Tyrer-Cuzick would be the in house lifestyle and familial risk factors model WITHOUT AI.

R5-7. As commented above, we disagree with this approach. We also point out that Tyrer-Cuzick v8 in our comparison includes mammographic density.

  1. Why have the authors not used TC along with their AI density model as a comparison?

R5-8. As stated above we do not see the purpose or reason to generate completely new models and compare the two.

Please present data on the models from 2-10 years excluding the next mammogram and interval cancers to 2 years

R5-9. We agree with the reviewer that such an analysis also would be of interest. However, since it is a completely new approach, we will not be able to fit it in to the current paper.

Round 2

Reviewer 4 Report

The authors addressed my concerns.

Author Response

Thanks. We see no more comments to respond to.

Reviewer 5 Report

The authors have partially responded to this reviewers concerns. There are still issues that have not been adequately addressed

1.       We do not see the purpose of making comparisons between non-existing models, where the AI-based model does not exist without the AI component and Tyrer-Cuzick does not exist with AI and how this is related to being internationally useful. We also like to clarify that the AI risk model is currently offered to clinics in U.S. and Europe.

Please make it clear how the model can be used by readers of this paper

2.         ‘The increased performances were observed in multiple risk subgroups and cancer subtypes. In the high-risk women per NICE guidelines, 22%, 19%, 7.2% of BCs were identified using the lifestyle/familial-expanded model, the imaging-only model, p<0.01, and Tyrer-Cuzick, p<0.01.’ -it is not appropriate to compare a risk model (Tyrer-Cuzick) that contains a density input with a mammography imaging model without at least imputing a basic mammographic density input such as BiRads. The only correct comparison is with Tyrer-Cuzick and the new lifestyle/familial component without AI

R5-3. We would like to clarify that Tyrer-Cuzick v8 includes mammographic density. In the introduction we have written that

The authors do not make it clear how they used includes mammographic density in Tyrer-Cuzick v8. This needs to be explicit in the methods section. How was density measured for the TC8 input?

3.       ‘TyrerCuzick provides breast cancer risk based on lifestyle and familial risk factors, i.e. age, height, weight, age at menarche, age at first childbirth, menopausal status, use of HRT, previous benign breast disorders, first and second-degree family history of breast cancer, first degree family history of ovarian breast cancer, and mammographic density categorized into BI-RADS categories [20].’ -This is incorrect. Breast cancer family history in cousins (third degree relatives) is also included. Ovarian cancer is also included for second degree relatives. Mammographic density is also possible using continuous measures such as visual assessment and Volpara. Please correct this. Please also stipulate whether BiRads was entered in the T-C model

R5-4. Thank you for pointing this out. We have now expanded section 2.2. in Methods and included:

We also clarify that we include BiRads density in the T-C model

You still have not corrected that Ovarian cancer is also included for second degree relatives, not that there is an input using continuous measures such as visual assessment and Volpara. Please correct this. You have also NOT clarified that you used BiRads in TC8. Please state how the BiRads score was derived explicitly. Was this from a single or agreed from two radiologists or derived from an automatic measure?

4.       Our comparison with Tyrer-Cuzick v8 was limited by the fact that we did not have information on ovarian cancer in second degree relatives.’

Please also add that using a continuous model such as visual assessment density with TC8 is likely to produce better results than BiRads as per PROCAS publications

5.       Why is table 2  now moved to supplementary? The 2.6% only in the 8% 10-year category still looked low for TC8 in control

6.       Please present data on the models from 2-10 years excluding the next mammogram and interval cancers to 2 years

R5-9. We agree with the reviewer that such an analysis also would be of interest. However, since it is a completely new approach, we will not be able to fit it in to the current paper.

I totally disagree. You have the data. The point is how good is the AI component at predicting risk after 2 years. Please do this analysis. You need to show that your model is not just better than a standard TC8+BiRads  for the first 2 years only

Author Response

The review comments for review round 2 are found in bold, our responses to these are in italic.

Reviewer 4

The authors addressed my concerns.

Reviewer 5

The authors have partially responded to this reviewers concerns. There are still issues that have not been adequately addressed

  1. We do not see the purpose of making comparisons between non-existing models, where the AI-based model does not exist without the AI component and Tyrer-Cuzick does not exist with AI and how this is related to being internationally useful. We also like to clarify that the AI risk model is currently offered to clinics in U.S. and Europe.

Please make it clear how the model can be used by readers of this paper

Response: We have now clarified in the introduction as follows:

“The AI-risk models are available for clinical use in the U.S. and Europe [18]”

  1. ‘The increased performances were observed in multiple risk subgroups and cancer subtypes. In the high-risk women per NICE guidelines, 22%, 19%, 7.2% of BCs were identified using the lifestyle/familial-expanded model, the imaging-only model, p<0.01, and Tyrer-Cuzick, p<0.01.’ -it is not appropriate to compare a risk model (Tyrer-Cuzick) that contains a density input with a mammography imaging model without at least imputing a basic mammographic density input such as BiRads. The only correct comparison is with Tyrer-Cuzick and the new lifestyle/familial component without AI

R5-3. We would like to clarify that Tyrer-Cuzick v8 includes mammographic density. In the introduction we have written that

The authors do not make it clear how they used includes mammographic density in Tyrer-Cuzick v8. This needs to be explicit in the methods section. How was density measured for the TC8 input?

Response: We used the automated mammographic density tool STRATUS (DOI:10.1007/s10549-018-4690-5) that lately also was used when the BOADICEA risk tool was expanded to include mammographic density (DOI: 10.1136/jmg-2022-108806). STRATUS generated a computerized BiRads classification that was added to the model. A recent report has also been published using STRATUS vs. Volpara for the same BOADICEA risk tool (DOI: 10.1158/1538-7445.AM2023-4174). We have now added the following text into the methods section:

“Tyrer-Cuzick v8 provides breast cancer risk based on lifestyle and familial risk factors, i.e. age, height, weight, age at menarche, age at first child-birth, menopausal status, use of hormone replacement therapy (HRT), previous benign breast disorders, first, second, and third-degree family history of breast cancer, first and second degree family history of ovarian breast cancer, and mammographic density. In our analysis, we did not have access to information regarding history of ovarian cancer in second degree relatives. We classified mammographic density using the fully automated mammographic density tool STRATUS [20]. Mammographic density was classified into four cBIRADS categories to mimic the Breast Imaging Reporting and Data System (BI-RADS) classification as previously described (22)”

  1. ‘TyrerCuzick provides breast cancer risk based on lifestyle and familial risk factors, i.e. age, height, weight, age at menarche, age at first childbirth, menopausal status, use of HRT, previous benign breast disorders, first and second-degree family history of breast cancer, first degree family history of ovarian breast cancer, and mammographic density categorized into BI-RADS categories [20].’ -This is incorrect. Breast cancer family history in cousins (third degree relatives) is also included. Ovarian cancer is also included for second degree relatives. Mammographic density is also possible using continuous measures such as visual assessment and Volpara. Please correct this. Please also stipulate whether BiRads was entered in the T-C model

R5-4. Thank you for pointing this out. We have now expanded section 2.2. in Methods and included:

We also clarify that we include BiRads density in the T-C model

 You still have not corrected that Ovarian cancer is also included for second degree relatives, not that there is an input using continuous measures such as visual assessment and Volpara. Please correct this. You have also NOT clarified that you used BiRads in TC8. Please state how the BiRads score was derived explicitly. Was this from a single or agreed from two radiologists or derived from an automatic measure?

Response: Please also see our response above.

  1. Our comparison with Tyrer-Cuzick v8 was limited by the fact that we did not have information on ovarian cancer in second degree relatives.’

Please also add that using a continuous model such as visual assessment density with TC8 is likely to produce better results than BiRads as per PROCAS publications

Response: We suggest adding a general statement that a continuous measure was not used to the limitation section:

“Our comparison with Tyrer-Cuzick v8 was limited by the fact that we did not have information on ovarian cancer in second degree relatives and, that we did not use a continuous measure for mammographic density.”

  1. Why is table 2  now moved to supplementary? The 2.6% only in the 8% 10-year category still looked low for TC8 in control

Response: In accordance with suggestions by reviewer 1, the table was moved to supplementary results. We would also like to refer to reports on how mammographic density, using our STRATUS tool, alters the risk distribution of the BOADICEA risk tool (DOI: 10.1136/jmg-2022-108806) and (DOI: 10.1158/1538-7445.AM2023-4174).

  1. Please present data on the models from 2-10 years excluding the next mammogram and interval cancers to 2 years

R5-9. We agree with the reviewer that such an analysis also would be of interest. However, since it is a completely new approach, we will not be able to fit it in to the current paper.

 I totally disagree. You have the data. The point is how good is the AI component at predicting risk after 2 years. Please do this analysis. You need to show that your model is not just better than a standard TC8+BiRads  for the first 2 years only

Response: We have now included the requested analysis as follows: In methods we added:

“In a subgroup analysis, we also excluded breast cancers diagnosed in the first two years after study baseline.”

In Results we also added the following:

“When excluding breast cancers diagnosed in the first two years after study baseline, we observed AUC point estimates for the lifestyle/familial-expanded AI-risk model ranging from 0.72 (95%CI 0.67-0.77) after 3-years of follow-up to 0.65 (95%CI 0.61-0.69) after 10-years of follow-up, Supplementary Figure A2. The corresponding estimates for the imaging-only AI-risk model were 0.70 (95%CI 0.65-0.75) and 0.61 (95%CI 0.57-0.65) and for Tyrer-Cuzick v8 model, 0.63 (95%CI 0.54-0.69) and 0.60 (95%CI 0.58-0.62).”

Round 3

Reviewer 5 Report

The authors have now responded to the reviewer comments. The manuscript is now much better and the inclusion of the 3-10 year analysis is important. They still show a significant advantage of their combined model albeit against a BiRads incorporated TC8